# Mumefural Improves Blood Flow in a Rat Model of FeCl_3_-Induced Arterial Thrombosis

**DOI:** 10.3390/nu12123795

**Published:** 2020-12-10

**Authors:** Jihye Bang, Won Kyung Jeon

**Affiliations:** 1Herbal Medicine Research Division, Korea Institute of Oriental Medicine, 1672 Yuseong-daero, Yuseong-gu, Daejeon 34054, Korea; jhbang0920@kiom.re.kr; 2Convergence Research Center for Diagnosis, Treatment and Care System of Dementia, Korea Institute of Science and Technology, 5 Hwarang-ro 14-gil, Seongbuk-gu, Seoul 02792, Korea

**Keywords:** mumefural, FeCl_3_-induced arterial thrombosis, platelet activation

## Abstract

Mumefural (MF), a bioactive component of the processed fruit of *Prunus mume* Sieb. et Zucc, is known to inhibit platelet aggregation induced by agonists in vitro. In this study, we investigated the anti-thrombotic effects of MF using a rat model of FeCl_3_-induced arterial thrombosis. Sprague–Dawley rats were intraperitoneally injected with MF (0.1, 1, or 10 mg/kg) 30 min before 35% FeCl_3_ treatment to measure the time to occlusion using a laser Doppler flowmeter and to assess the weight of the blood vessels containing thrombus. MF treatment significantly improved blood flow by inhibiting occlusion and thrombus formation. MF also prevented collagen fiber damage in injured vessels and inhibited the expression of the platelet activation-related proteins P-selectin and E-selectin. Moreover, MF significantly reduced the increased inflammatory signal of nuclear factor (NF)-κB, toll-like receptor 4 (TLR4), tumor necrosis factor (TNF)-α, and interleukin (IL)-6 in blood vessels. After administration, MF was detected in the plasma samples of rats with a bioavailability of 36.95%. Therefore, we suggest that MF may improve blood flow as a candidate component in dietary supplements for improving blood flow and preventing blood circulation disorders.

## 1. Introduction

Blood flow is determined by the action of the cardio-cerebrovascular system and blood fluidity, and reduced blood flow can cause various serious problems [1,2]. In injured blood vessels, platelet activation causes thrombus formation and reduces blood flow through complex mechanisms involving interactions of various receptors and ligands [3,4,5]. Many studies have reported that the inhibition of platelet activation or aggregation may be useful in preventing thrombosis [6,7].

Dietary supplements such as ginkgo biloba, resveratrol, quercetin, curcumin, catechin, and omega-3 fatty acids are known to inhibit platelet activation [6,7,8,9]. In recent decades, *Prunus mume* fruit extracts (PMFE) has been reported to have various health benefits [10,11,12,13], including improvements of blood flow [14] and inhibition of platelet aggregation [15]. We have previously reported that PMFE improves cognitive impairment induced by decreased blood flow through an inhibitory effect on the toll-like receptor 4(TLR4)/myeloid differentiation primary response 88/ nuclear factor (NF)-κB signaling pathways [16,17]. According to Nishimura et al., PMFE containing 1% mumefural (MF; 2-{2-[(5-Formylfuran-2-yl)methoxy]-2-oxoethyl}-2-hydroxybutanedioic acid) improves blood pressure in hypertensive patients after daily intake for 12 weeks [18].

MF is a citric acid derivative produced by heat processing of *Prunus mume* (*P. mume*) fruit or lemon juice [18,19,20]. MF is an effective component of PMFE, which is known to inhibit platelet aggregation [15] and promote erythrocyte deformability [20]. *P. mume* fruit containing MF is widely used as a health food, especially in Japan [18,19,20]. We have previously shown that MF ameliorates cognitive impairment resulting from reduced cerebral blood flow by modulating the cholinergic system and neuroinflammation [21]. Although there are few studies on the efficacy of MF to date, it is expected to have an excellent effect on inhibiting platelet activity and suppressing inflammation.

In this study, we investigated the anti-thrombotic efficacy of MF by assessing the inhibition of platelet activation and reduction of inflammation in an FeCl_3_-induced arterial thrombosis rat model.

## 2. Materials and Methods

### 2.1. Animals

Seven-week-old Sprague–Dawley (SD) rats were purchased from Orient Bio (Seongnam, Korea). The animals were housed under a 12 h light/dark cycle at 22 ± 1 °C, with a relative humidity of 55 ± 10%, and they were fed water and chow ad libitum. The experimental procedures were approved (KIST-2020-023) by the Institutional Animal Care and Use Committee of the Korean Institute of Science and Technology (Seoul, Korea).

### 2.2. FeCl_3_-Induced Vascular Injury and Drug Treatment

After 1 week of adaptation, the rats were randomly assigned to a total of 9 groups; 6 groups of varying drug dose by intraperitoneal injection (Control, FeCl_3_, 0.1, 1, 10 mg/kg of MF, and 10 mg/kg of Gingkolide (GA)) and 3 groups by oral administration (Control, FeCl_3_, 10 mg/kg of MF). MF with a purity > 95%, analyzed using HPLC, was dissolved in saline before the experiments. Gingkolide (GA), the major component of *Ginkgo biloba* extracts [22], was used as a positive Control at 10 mg/kg and purchased from Sigma Aldrich (St. Louis, MO, USA). Doses of 0.1, 1, and 10 mg/kg of MF, and 10 mg/kg of GA were administered intraperitoneally at once 30 min before FeCl_3_-induced vascular injury, while 10 mg/kg of MF was administered by oral gavage once a day for 1 week. The Control and FeCl_3_-induced groups were administered the same amount of saline. In the Control group, the carotid artery was not exposed to FeCl_3_. Rats were anesthetized with isoflurane (Hana Pharm., Kyonggi-do, Korea) for the operative procedures. The right of the carotid artery was exposed and treated with 35% FeCl_3_-saturated filter paper (0.2 cm × 0.2 cm) for 3 min as previously described [23]. Next, the filter paper was removed and blood flow was measured using a laser Doppler flowmeter (LDF; BFL21, Transonic Instrument, Ithaca, NY, USA) until occlusion of the exposed carotid artery. Analysis of vascular occlusion time was performed using the chart4 program (Analogue Digital Instruments Inc, Victoria, Australia). The FeCl_3_-treated carotid artery was cut off, dried on filter paper, and weighed.

### 2.3. Immunostaining

Animals were sacrificed after the experiments, and blood vessel tissues were collected for analysis. Subsequently, the carotid arteries were fixed in formaldehyde, embedded in paraffin, 4-μm sectioned, and deparaffinized. The sections were rehydrated and stained using hematoxylin and eosin (H&E) and Masson’s trichrome (MT) staining according to the manufacturer’s protocols (Abcam, MA, USA). The collagen fiber measurement was quantified using Meta Image Series 7.7 (Molecular Devices, PA, USA), using an indirect method that measures the ratio of the area occupied by the collagen fiber in the blood vessel to the area of entire blood vessel tissue. Immunofluorescence analyses of P-selectin, E-selectin, platelet/endothelial cell adhesion molecule-1 (PECAM-1), intercellular adhesion molecules (ICAM), vascular cell adhesion molecule (VCAM), NF-κB, tumor necrosis factor (TNF)-α, interleukin (IL)-6, and TLR4 were performed. Primary antibody information is presented in Table 1. Carotid arteries were prepared and incubated with primary antibodies in PBS containing 5% horse serum and 0.1% Triton-X 100 overnight at 4 °C. After washing with PBS, the tissues were incubated with secondary antibodies (Cell Signaling, Danvers, MA, USA) and mounted onto resin-coated slides with Immu-Mount reagent (Thermo Scientific, Pittsburgh, PA, USA). All immunoreactions were examined using fluorescence microscopy (ECLIPs; Nikon, Tokyo, Japan). Densitometric analysis was performed using Image J (NIH, Bethesda, MD, USA) and densities were calculated as follows: density of target molecule (%) = FeCl_3_ or drug treatment group / Control × 100 (%). A minimum of three sections was selected for each rat, and the results were averaged for analysis.

### 2.4. Pharmacokinetics Analysis

Pharmacokinetic (PK) analysis and interpretation of the results were performed by Medicilon Preclinical Research (Shanghai, China) LLC under performed in non-GLP conditions. The PK parameters were determined by the study director for the test article from individual concentration–time data in the test species. Phoenix WinNonlin^®^ 7.0 (Pharsight Corporation, Mountain View, CA, USA) was used to calculate the PK parameters. Any concentrations that were below the lower limit of quantification (BLQ) were omitted from the calculation of PK parameters.

### 2.5. Statistical Analysis

Results are expressed as the mean ± standard deviation (SD). All data were analyzed using one-way repeated analysis of variance (ANOVA), followed by LSD post hoc tests. A *p*-value of < 0.05 was considered statistically significant. All statistical analyses were performed using SPSS 20.0 Software (IBM, Chicago, IL, USA).

## 3. Results

### 3.1. MF Improves Blood Flow in an FeCl_3_-Induced Araterial Thrombosis Model

We determined the effects of MF on vascular occlusion time after FeCl_3_-induced artery thrombosis. The mean time to occlusion (TTO) for 1 mg/kg and 10 mg/kg MF (21.38 ± 1.13, and 29.14 ± 0.42, respectively) was significantly longer than that in the FeCl_3_ group (9.28 ± 0.15 min; *p* < 0.001, Figure 1A). In addition, the positive control drug GA increased the TTO ratio by 3.08 ± 0.07 times compared to that in the FeCl_3_ group (*p* < 0.001). Treatment with 1, 10, and 10 mg/kg GA markedly reduced blood vessel weight (0.56 ± 0.02, 0.53 ± 0.02, and 0.55 ± 0.01 mg/mm, respectively) compared with vehicle (0.83 ± 0.03 mg/mm; *p* < 0.01, Figure 1B). However, treatment with 0.1 mg/kg MF did not change the TTO or blood vessel weight compared with FeCl_3_. In addition, when 10 mg/kg MF was administered orally, the mean TTO (29.00 ± 3.37 min) was delayed compared to that in the FeCl_3_ group (9.59 ± 0.32 min; *p* < 0.001, Figure 1C). After oral administration of 10 mg/kg MF once a day for a week, blood vessel weight decreased in the MF-administered group (0.53 ± 0.01 mg/mm) compared to that in the FeCl_3_ group (0.83 ± 0.05 mg/mm; *p* < 0.001, Figure 1D). Taken together, these results suggest that MF is effective in improving blood flow disorders resulting from FeCl_3_-induced vascular injury, and that MF efficacy was not affected by the administration method.

### 3.2. MF Inhibits Collagen Fiber Damage in an FeCl_3_-Induced Araterial Thrombosis Model

FeCl_3_ induces vascular injuries, such as endothelial denudation, collagen fiber damage, and thrombus formation. H&E staining was performed to investigate histological differences between the Control and FeCl_3_-induced vascular injury, and Masson’s trichrome staining was used to quantify areas containing collagen fibers. Figure 2A shows representative photomicrographs of H&E or Masson’s trichrome staining in the carotid artery after treatment with FeCl_3_. Blood coagulation, thrombus, and platelet activation were observed after FeCl_3_ treatment, and MF and GA inhibited these effects. In addition, collagen fiber damage was detected in FeCl_3_-treated rats (11.58 ± 5.34%, Figure 2B), but it was significantly ameliorated after 10 mg/kg MF (23.23 ± 5.67%, *p* < 0.001) and 10 mg/kg GA (24.56 ± 4.63%, *p* < 0.001) treatments compared with that in the FeCl_3_ group (11.58 ± 5.34%). In a previous study, collagen fiber damage was not observed in the carotid artery of Control group rats. Treatment with 10 mg/kg MF and 10 mg/kg GA restored FeCl_3_-induced collagen fiber damage by more than 2-fold. Unfortunately, the protective effect on collagen fibers was not statistically significant after treatment with 0.1 and 1 mg/kg MF. These data suggest that 10 mg/kg MF may be an effective concentration to restore the damage to collagen fibers induced by FeCl_3_ treatment.

### 3.3. MF Inhibits Cell Adhesion Molecules (CAMs) in an FeCl_3_-Induced Arterial Thrombosis Model

After thrombosis induction by FeCl_3_, platelets rapidly adhered around blood clots and thrombi. Furthermore, after treatment with FeCl_3_, the expression of CAMs increased in blood vessels. CAMs including PECAM-1, P-selectin, E-selectin, ICAM, and VCAM expression levels were all significantly higher in arterial tissues induced by FeCl_3_ treatment (*p* < 0.001; Figure 3A–E, respectively). Figure 3A shows that MF treatment at 10 mg/kg reduced the damage to the carotid artery vascular tissue, the optical density of platelet activation, and the relative regional PECAM-1 staining after FeCl_3_ treatment (*p* < 0.001). Figure 3B shows that MF treatment at 1 and 10 mg/kg significantly reduced P-selectin related to platelet activation in the carotid artery vascular tissue compared to that in the untreated FeCl_3_ group (FeCl_3_ + MF 1 mg/kg, *p* < 0.01; FeCl_3_ + MF 10 mg/kg, *p* < 0.001). Figure 3C shows that MF treatment at 1 and 10 mg/kg reduced the damage to the carotid artery vascular tissue, the optical density of platelet activation, and the relative regional E-selectin^+^ staining after FeCl_3_ treatment (FeCl_3_ + MF 1 mg/kg, *p* < 0.01; FeCl_3_ + MF 10 mg/kg, *p* < 0.001; Figure 3C). GA had a platelet activation inhibitory effect similar to that of 10 mg/kg MF.

ICAM staining of the FeCl_3_ group increased sharply compared to that of the Control group (*p* < 0.001, Figure 3D). MF at 10 mg/kg significantly reduced the expression of ICAM in the carotid artery compared to the FeCl_3_ group (10 mg/kg MF, *p* < 0.001), and VCAM staining of the FeCl_3_ group increased sharply compared to that of the Control group (*p* < 0.001, Figure 3E). MF at 1 and 10 mg/kg significantly reduced the expression of VCAMs in the carotid artery compared to the FeCl_3_ group (1 mg/kg MF, *p* < 0.01; 10 mg/kg MF, *p* < 0.001). In addition, 10 mg/kg MF decreased VCAM staining compared to the FeCl_3_ group (*p* < 0.001) and showed similar efficacy as FeCl_3_ + GA 10 mg/kg (*p* < 0.05). These data suggest that MF inhibits platelet activation induced by FeCl_3_ treatment and that MF inhibits the expression of CAMs induced by FeCl_3_ treatment.

### 3.4. MF Inhibits Inflammatory Response in an FeCl_3_-Induced Arterial Thrombosis Model

Molecules related to platelet activation are induced by inflammatory reactions. Platelet activation and inflammation caused by vascular injury are closely related. NF-κB^+^ and TNF-α^+^ staining was increased in the FeCl_3_ group increased compared to that of the Control group (*p* < 0.001, Figure 4A). The expression levels of NF-κB^+^ and TNF-α^+^ in the MF-treated groups were significantly reduced compared to those in the FeCl_3_ group in a MF concentration-dependent manner (0.1 and 1 mg/kg MF, *p* < 0.01; 10 mg/kg MF, *p* < 0.001). In addition, 10 mg/kg MF showed a better anti-inflammatory activity than did FeCl_3_ + GA (10 mg/kg, *p* < 0.05). TLR4^+^ and IL-6^+^ in the MF-treated groups were higher than those in the Control group (*p* < 0.001, Figure 4B). The expression levels of TLR4^+^ and IL-6^+^ were significantly reduced compared to those in the FeCl_3_ group in a MF concentration-dependent manner (0.1 and 1 mg/kg MF, *p* < 0.01; 10 mg/kg MF, *p* < 0.001). Moreover, 10 mg/kg MF showed a better anti-inflammatory activity than did FeCl_3_ + GA (10 mg/kg, *p* < 0.05). These data suggest that MF has excellent efficacy in inhibiting inflammation induced by FeCl_3_ treatment.

### 3.5. Pharmacokinetic Parameters of MF in an FeCl_3_-Induced Arterial Thrombosis Model

The plasma concentration–time curves are shown in Figure 5. Plasma concentration and plasma PK parameters for individual animals are shown in Table 2 and Table 3, respectively. Table 2 shows blood MF concentration over time in individual animals. Concentrations below 10 ng/mL are indicated as BLQ. Under the experimental conditions, SD rats intravenously (IV) injected with MF at 2 mg/kg showed a mean C_max_ of 1894.5 ng/mL and a mean AUC_(0–t)_ of 564.73 h*ng/mL, and SD rats with oral administration (PO) of MF at 10 mg/kg showed a mean C_max_ of 1731.61 ng/mL and a mean AUC_(0–t)_ of 1043.28 h*ng/mL. Therefore, the mean bioavailability of MF in SD rats was 36.95%. These data indicate that MF enters the blood and acts on the absorption–metabolism–excretion process.

## 4. Discussion

MF, a component of Japanese apricot juice concentrate PMFE [17,23,24], is well known to inhibit platelet aggregation [18], improve cognitive impairment [21], and have various beneficial effects [20,25,26,27]. In this study, we showed the anti-thrombotic efficacy and anti-inflammatory effects of MF in an FeCl_3_-induced thrombosis model.

Measuring blood flow in an FeCl_3_-induced thrombosis animal model is the most effective approach to be used when evaluating whether the model is well formed or whether the drug is effective [24,28]. The weight of the affected blood vessel is a representative measurement used for verifying the anti-thrombotic efficacy in FeCl_3_-induced animal models [23]. The FeCl_3_-exposed carotid artery is heavier due to the intravascular thrombus formation, blood coagulation, platelet aggregation, and leukocyte recruitment. Our results showed that time to occlusion delayed and blood vessel weight decreased in an MF concentration-dependent manner. We also found that MF delayed blood vessel occlusion by inhibiting thrombus formation. In addition, the intraperitoneal or oral administrations of MF also delayed time to occlusion and decreased vessel weight, including thrombus formation. This indicates that the efficacy of MF in the FeCl_3_-induced animal model was not affected by the administration method.

Establishment of FeCl_3_-induced vascular injury models involves the application of an FeCl_3_-soaked filter paper to the surface of an artery, which induces a free radical-based injury transferred to the luminal surface, disrupting the endothelium and inducing a thrombotic response [29,30]. Thus, thrombotic events and damaged collagen fibers were observed in the FeCl_3_-treated groups. Histological analysis of the FeCl_3_-treated carotid arteries showed complete occlusion [23]. However, FeCl_3_-induced occlusion was suppressed, and the damaged collagen fibers were ameliorated in the MF-treated groups compared to those in the FeCl_3_ group. We also found that MF inhibited blood vessel injury when thrombus was generated depending on the damage of collagen fibers by FeCl_3_ treatment.

Considering that the experimental conditions were suitable for observing the anti-thrombotic efficacy through local vascular changes, we judged that it was reasonable to analyze the expression of CAMs and inflammatory markers not only in endothelial cells but also in the entire blood vessel. PECAM-1, one of CAMs, was used as a platelet endothelial cell adhesion molecule, and it is present in endothelial cells as well as platelets, macrophages, lymphocytes, osteoclasts, and other cells [31,32]. Consequently, not only endothelial cells but also platelets and thrombi generated in blood vessels were stained with PECAM-1 in the blood vessels damaged by FeCl_3_ (Figure 3A). Selectins are molecules related to platelet activation in thrombus generation. During acute and chronic inflammatory reactions, P-selectin is first expressed, which is followed by the expression of E-selectin [33,34,35]. Selectin appears to be important for thrombus formation and inflammatory enhancement, whereas IL-10 appears to inhibit these processes [36,37]. Inflammatory cytokines such as TNF-α, up-regulate the endothelial synthesis of E-selectin [38]. In particular, leucocytes, mostly neutrophils, rapidly roll on mobilized P-selectin [39]. In this study, the distribution of selectin in vascular tissues involved in FeCl_3_-induced thrombus formation was measured by immunofluorescence staining. MF reduced the expression of P-selectin and E-selectin compared with the levels in the FeCl_3_ treatment group. Accordingly, MF inhibited the platelet activation and CAMs expression, which were activated by FeCl_3_-induced thrombus formation, thereby reducing thrombus formation, delaying blood vessel occlusion, and improving blood flow.

A short exposure to FeCl_3_ results in an acute response that causes severe vascular damage, reactive oxygen species production, and inflammatory reactions in blood vessels. It is known that the onset of inflammation increases the activity of CAMs such as ICAM or VCAM, and it increases the adhesion of circulating monocytes and leukocytes to the vascular endothelium [40]. Our results showed that MF reversed the increased expression of ICAM or VCAM induced by FeCl_3_ treatment and inhibited thrombus formation. These data suggest that MF reduces the activity of ICAM and VCAM by inhibiting acute inflammation.

Especially, TNF-α, one of the most potent pro-inflammatory cytokines, is critically implicated not only in the induction of endothelial apoptosis but also in development [41]. NF-κB plays an important role in the transcriptional regulation of inflammatory proteins [42]. Furthermore, TLR4 is a transmembrane receptor known to activate NF-κB and IL-6 release when stimulated by cytokines such as TNF-α and lipopolysaccharide [43,44]. Interestingly, in this study, MF reversed the increased expression of inflammation-related molecules induced by FeCl_3_ treatment. In particular, MF showed a statistically significantly higher anti-inflammatory effect than GA did. As a result, MF strongly inhibited the inflammatory response induced by FeCl_3_, thereby inhibiting the activity of selectins and CAMs, delaying the formation of thrombus, and improving blood flow.

PK analysis, as reflected by C_max_ (IV: 1594.54 ± 580.66 ng/mL, PO: 1731.61 ± 290.64 ng/mL) and T_max_ (IV: 1 h, PO: 4 h), revealed MF to be present in the plasma after each single administration (IV: 2 mg/kg; oral: 10 mg/kg). Hence, the bioavailability of MF was found to be 36.95%. These data can be used to predict the effects of MF in clinical trials or in vivo studies.

Taken together, our results show that MF improves blood flow in an FeCl_3_-induced thrombosis model by inhibiting platelet activation and suppressing inflammation, suggesting that MF can be developed as a candidate dietary supplement for preventing the onset of vascular diseases caused by blood flow disorders.

## Figures and Tables

**Figure 1 nutrients-12-03795-f001:**
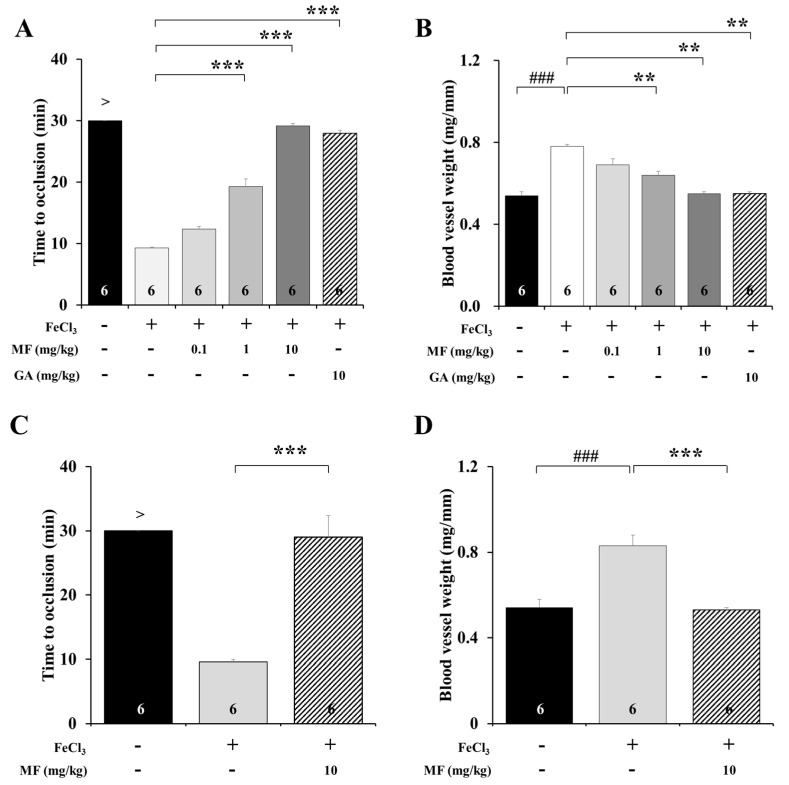
Time to occlusion (**A**) and blood vessel weight (**B**) in an FeCl_3_-induced arterial thrombosis model with an intraperitoneal injection of MF. GA was used as a positive Control. Time to occlusion (**C**) and blood vessel weight (**D**) in FeCl_3_-induced arterial thrombosis model with oral injection of MF. Data are expressed as mean ± standard deviation. ### *p* < 0.001, compared with the Control group; ** *p* < 0.01 and *** *p* < 0.001, compared with the FeCl_3_ group; *n* = 6 rats per group. MF: mumefural; GA: gingkolide A.

**Figure 2 nutrients-12-03795-f002:**
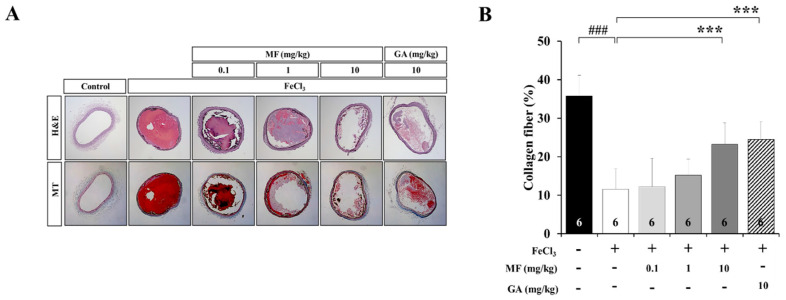
Histological examination of FeCl_3_-induced carotid artery thrombosis after MF treatment. Representative images of H&E and MT staining of vessels (**A**) and blue-stained collagen fibers (**B**), showing that the effects of MF treatment on an FeCl_3_-induced arterial thrombosis model. GA was used as a positive Control drug. Data are expressed as mean ± standard deviation. ### *p* < 0.001, compared with the Control group; *** *p* < 0.001, compared with the FeCl_3_ group; *n* = 6 rats per group. MF: mumefural; GA: gingkolide A; H&E: hematoxylin and eosin; MT: Masson’s trichrome.

**Figure 3 nutrients-12-03795-f003:**
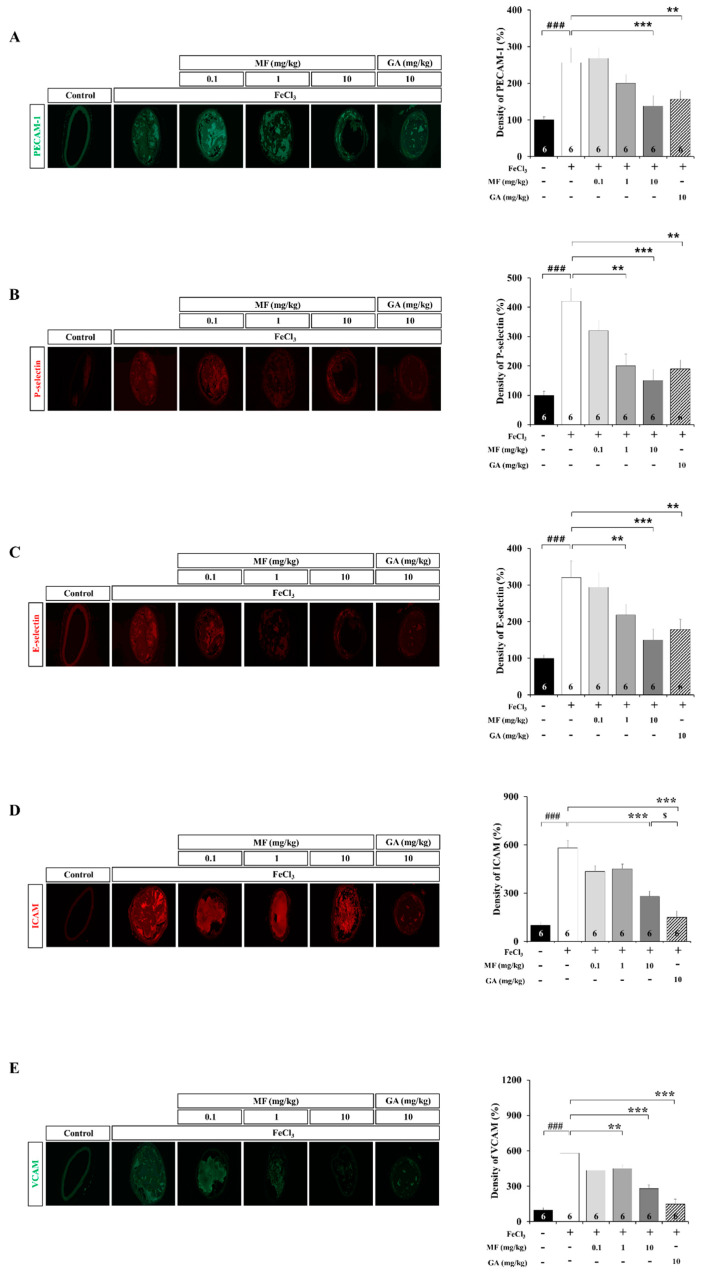
Cell adhesion molecules in an FeCl_3_-induced arterial thrombosis model. Images show immunofluorescence staining for PECAM-1 (green) (**A**), P-selectin (red) (**B**), E-selectin (red) (**C**), ICAM (red) (**D**), and VCAM (green) (**E**). GA was used as a positive Control drug. Data are expressed as mean ± standard deviation. ### *p* < 0.001, compared with the Control group; ** *p* < 0.01 and *** *p* < 0.001, compared with the FeCl_3_ group; $ *p* < 0.05, compared with the FeCl_3_+MF 10 mg/kg group; *n* = 6 rats per group. MF: mumefural; GA: gingkolide A; PECAM-1: platelet endothelial cell adhesion molecule; ICAM: intercellular adhesion molecule; VCAM: vascular cell adhesion molecule.

**Figure 4 nutrients-12-03795-f004:**
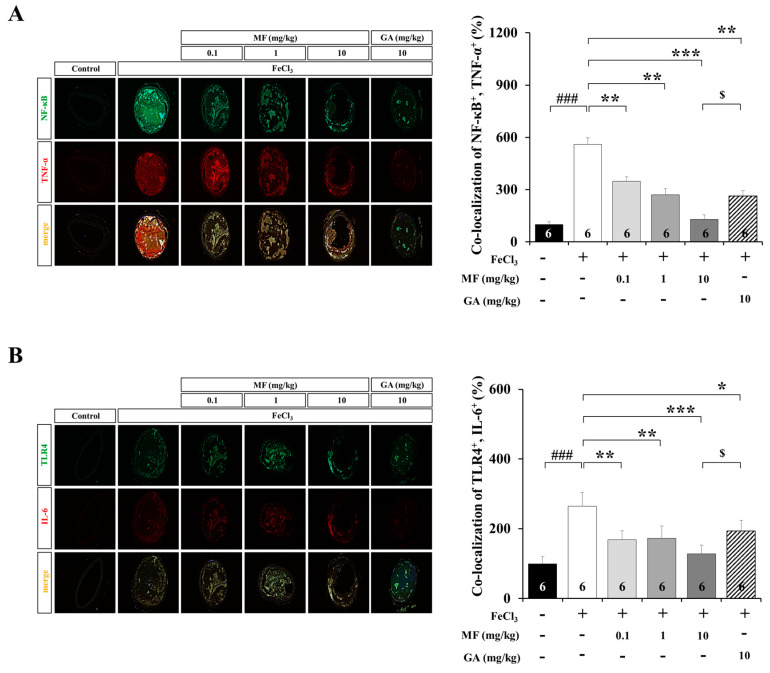
Expression of inflammation-related markers in an FeCl_3_-induced arterial thrombosis model. Images show immunofluorescence staining for NF-κB (green) and TNF-α (red) with inflammatory signaling marker and inflammatory cytokine co-localization indicated by yellow (merge) (**A**). Images show immunofluorescence staining for TLR4 (green) and IL-6 (red) with inflammatory signaling marker and inflammatory cytokine co-localization indicated by yellow (merge) (**B**). GA was used as a positive Control drug. Data are expressed as mean ± standard deviation. ### *p* < 0.001, compared with the Control group; * *p* < 0.05, ** *p* < 0.01 and *** *p* < 0.001, compared with the FeCl_3_ group; $ *p* < 0.05, compared with the FeCl_3_ + MF 10 mg/kg group; *n* = 6 rats per group. MF: mumefural; GA: gingkolide A; NF-κB: nuclear factor-κB; TNF-α: tumor necrosis factor-α; TLR4: toll-like receptor4; IL-6: interleukin-6.

**Figure 5 nutrients-12-03795-f005:**
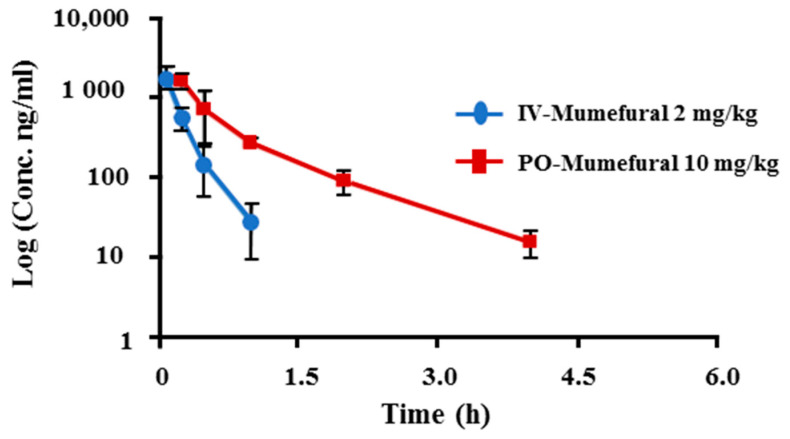
Mean plasma concentration-time curve of MF in Sprague–Dawley rats following intravenous injection and oral administration. Data are expressed as mean ± standard deviation from four rats at different time points. IV: intravenous injection; PO: oral administration.

**Table 1 nutrients-12-03795-t001:** Primary antibodies used in the experiments.

Primary Antibody	Company	Dilution
Anti-P-selectin	Thermo Scientific	1:250
Anti-E-selectin	Abcam	1:250
Anti-platelet/endothelial cell adhesion molecule-1 (PECAM-1)	Abcam	1:100
Anti-intercellular Adhesion Molecule 1 (ICAM)	Abcam	1:500
Anti-vascular cell adhesion molecule (VCAM)	Abcam	1:500
Anti-nuclear factor (NF)-κB	Santa Cruz	1:1000
Anti-tumor necrosis factor (TNF)-α	Santa Cruz	1:500
Anti-interleukin (IL)-6	Santa Cruz	1:500
Anti-Toll like receptor 4 (TLR4)	Santa Cruz	1:1000

**Table 2 nutrients-12-03795-t002:** Plasma MF concentration in SD rats.

**IV MF (2 mg/kg)**						
**Blood concentration (ng/mL)**						
**Time (h)**	**Rat 1**	**Rat 2**	**Rat 3**	**Rat 4**	**Mean**	**SD**
0	0.00	0.00	0.00	0.00	0.00	0.00
0.083	1635.95	1329.02	1928.95	2684.24	1894.54	580.66
0.25	826.55	434.42	429.42	658.24	587.16	191.98
0.5	197.32	84.42	67.56	272.14	155.36	96.85
1	23.08	13.16	BLQ	49.44	28.56	18.75
2	BLQ	BLQ	BLQ	BLQ	NA	NA
4	BLQ	BLQ	BLQ	BLQ	NA	NA
6	BLQ	BLQ	BLQ	BLQ	NA	NA
8	BLQ	BLQ	BLQ	BLQ	NA	NA
24	BLQ	BLQ	BLQ	BLQ	NA	NA
**PO MF (10 mg/kg)**						
**Blood concentration (ng/mL)**						
**Time (h)**	**Rat 1**	**Rat 2**	**Rat 3**	**Rat 4**	**Mean**	**SD**
0	0.00	0.00	0.00	0.00	0.00	0.00
0.25	2107.84	1268.53	1648.32	1759.75	1696.11	345.77
0.5	432.06	1410.53	424.37	686.58	738.38	464.36
1	272.87	303.51	232.32	306.84	278.89	34.60
2	55.41	121.98	85.24	116.78	94.85	30.90
4	21.39	9.52	BLQ	18.01	16.31	6.12
6	BLQ	BLQ	BLQ	BLQ	NA	NA
8	BLQ	BLQ	BLQ	BLQ	NA	NA
10	BLQ	BLQ	BLQ	BLQ	NA	NA
24	BLQ	BLQ	BLQ	BLQ	NA	NA

Animals were administered MF either by IV (2 mg/kg) or PO (10 mg/kg). Blood MF concentrations were then measured at different time points. MF: mumefural; BLQ: below limit of quantitation; NA: not applicable; IV: intravenous injection; PO: oral administration; SD: standard deviation.

**Table 3 nutrients-12-03795-t003:** Plasma pharmacokinetic parameters of MF in SD rats.

	t_1/2_ (h)	T_max_ (h)	C_max_ (ng/mL)	AUC_(0–t)_ (h*ng/mL)	F (%)
**IV Mumefural 2 mg/kg**					
**Rat1**	0.15	0.083	1635.95	551.91	-
**Rat 2**	0.14	0.083	1329.02	387.8	-
**Rat 3**	0.09	0.083	1928.95	508	-
**Rat 4**	0.2	0.083	2684.24	811.19	-
**Mean**	0.14	0.08	1894.54	564.73	-
**SD**	0.05	0	580.66	178.35	-
**PO Mumefural 10 mg/kg**					
**Rat 1**	0.8	0.25	2107.84	998.14	35.35
**Rat 2**	0.59	0.5	1410.53	1266.2	44.84
**Rat 3**	0.65	0.25	1648.32	788.08	27.91
**Rat 4**	0.73	0.25	1759.75	1120.72	39.69
**Mean**	0.69	0.31	1731.61	1043.28	36.95
**SD**	0.69	0.13	290.64	202.37	7.17

AUC_(0–t)_: area under the curve from the time of dosing to infinity; C_max_: maximum concentration; F: bioavailability; T_1/2_: terminal half-life; T_max_: time of the maximum concentration; SD: standard deviation; F was calculated using the following formula
$$F=\left(AUC_{0-t\left(PO\right)}\times Dose_{\left(IV\right)}÷AUC_{0-t\left(IV\right)}\times Dose_{\left(PO\right)}\right)\times 100\%$$

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
