# Peer review of "Mumefural Improves Blood Flow in a Rat Model of FeCl3-Induced Arterial Thrombosis"

_nutrients, 2020, doi:10.3390/nu12123795_

Round 1
Reviewer 1 Report
The manuscript by Bang J. and Jeon W.K. presented the effect of Mumefural (MF) a bioactive component of prunus mume fruit known to inhibit platelet aggregation and improve blood fluidity. Experiments performed on a rat model of arterial thrombosis showed beneficial effects of the biocomponent. MF inhibited thrombus formation and prevented collagen fiber damage in relation to decreased expression of platelet activation related proteins and to reduction of inflammatory markers increase NF-kB, TLR4, TNF-alpha and IL-6.
This is an original study. However, the methods are not clearly presented and do not provide an understanding of the study design and data. Additionally, some results are inconsistent and do not clearly supported the conclusion.
Major points:
1- The methods are not clearly described which makes study and results difficult to analyse.
- Especially in the study design, the authors wrote: …rats were randomly divided into six groups as follows:.. and only five groups are presented (lines 55-56). More, in figure 1, we discovered 9 groups which are as follows: CTL, FeCl3, FeCl3+0.1MF, FeCl3+1MF, FeCl3+10MF, FeCl3+10GA, with oral administration and CTL, FeCl3 and FeCl3+10MF with intraperitoneal injection.
- The authors wrote: One of the carotid artery was exposed and treated with FeCl3 (line 63). Which one? Right or left? Always the same? Moreover, it is not clear if the authors exposed carotid artery to FeCl3 in all experiments or just for blood flow measurement. If FeCl3 exposition was performed for all experiments, the authors should take the non-treated carotid as control. Then, the FeCl3 treated artery was dried and weighed and vessel weight was different between groups. What is the physiological explanation of these differences, the authors did not discuss that result.
- How did the authors quantify collagen fiber and against which reference is the ratio in figure 2 calculated.
2- Immuno histological analysis and its conclusion are not convincing.
- Indeed, based on increased immuno labelling the authors said that E-selectin, P-selectin, NK-kB, TNF-alpha, expression was higher after FeCl3 treatment. I think that the increased labelling was related to thrombus, increased platelets and blood levels in artery and did not reflect increased expression level. Mumefural treatment reduced thrombosis formation in carotid artery and consequently, labelling is reduced. Expression levels should be investigated by western blots on blood samples and arterial tissue. It is the same comment for inflammatory markers.
- Moreover, immunohistological images are too small and cannot be analysed. Higher magnifications are recommended. CD-31 antibody labels endothelial cells and normally fluorescence is localized on the inner surface of arteries. Such profile could not be observed on images in figure 3. DAPI images are not necessary.
- In figure 4A, merge images corresponded to double labelling with ICAM and P-selectin (A) and VCAM and CD-31? (B). Instead of DAPI show P-selectin and CD-31 labelling.
- In figure 3 and 5 graphs represented co-localization while in figure 3 the graph represented localization . Please explain the difference and how did these data were obtained.
3- The pharmacokinetic study has not real relevance in that form especially as the two conditions tested do not correspond to in vitro study. Intraperitoneal injection in section 2.2 and intravenous injection in section 2.4.
Table 4: What do these parameters mean? Please describe and discuss all of them. How were they obtained and especially AUC? Normally, AUC needs to be compared on the same time scale.
Author Response
1.
Thank you for your recommendations. We have highlighted the changes made in the revised manuscript in green. First, we deleted the description of the groups from line 57 as the number of groups used for each assay is different. Second, we indicated that the right carotid artery was used in line 65 (Materials and Methods 2.2). The control group was administered the same amount of saline used to dissolve MF as a vehicle, and it was not exposed to FeCl3. Saline administration did not affect blood flow. Also, the following description of the control group was added to line 63 (Materials and Methods 2.2): “In the control group, the carotid artery was not exposed to FeCl3.” And, the weight of the blood vessels is a representative measurement for verifying the antithrombotic efficacy in FeCl3 animal models. In fact, the weight of blood clots generated would be the most appropriate measurement. However, this is technically challenging. Therefore, measuring the blood vessel changes due to FeCl3-induced damage is an indirect way to determine the antithrombotic efficacy. The reason that the carotid artery exposed to FeCl3 is heavier than the normal group is due to thrombus formation, blood coagulation, platelet aggregation, and leukocyte recruitment as a result of the FeCl3-induced vessel damage. As described above, blood vessel weight measurement is indirect, so changes in blood vessels are directly confirmed through H&E staining. This explanation was added to lines 234-237 (Discussion). Third, regarding the quantification of the collagen fiber, we have published a number of papers on the efficacy of anti-thrombosis agents using this method over the past several years. We have added the following description to lines 70-72 (Materials and Methods 2.2): “The collagen fiber measurement was quantified using Meta Image Series 7.7 (Molecular Devices, USA), using an indirect method that measures the ratio of the area occupied by the collagen fiber in the blood vessel and the area of the entire blood vessel tissue.”
2.
First, as can be seen in a number of papers (References 23, 28, 29, and 30), in the model using FeCl3, thrombus is the result of the inflammatory response, ROS production, platelet activation, etc., that occur in blood vessels within minutes after FeCl3 exposure. Our study reproduces these findings. We attempted several times to do western blot analysis using vessel tissues. However, the concentration of the extracted protein was low and band detection was not possible. Therefore, the expression was confirmed in the vascular tissue directly with immunofluorescence. Unfortunately, we did not consider using a blood sample in the original experimental plan due to the focus on the expression in the vascular tissue. Therefore, the suggested experiments cannot be performed. We plan to measure the level of inflammatory markers in blood samples in our future studies.
Second, we have deleted the DAPI images from Figures 3, 4, and 5. We used a polyclonal CD31 antibody that detects endothelial cells. However, when observing the expression of CD31 in the model used in this experiment, it was not specific, and it was confirmed that all blood vessels, intravascular platelets, and even thrombi were stained. CD31 is also expressed in various cells such as platelets, macrophages, and immune cells. Therefore, we believe that the CD31 antibody stained not only endothelial cells, but also other parts of blood vessels that underwent vascular damage with FeCl3. We observed the same pattern with the E-selectin, P-selectin, ICAM, and VCAM markers. We expected a change in the expression of E-/P-selectin in luminal lining endothelial cells. However, we judged that it would be more accurate to observe the changes occurring in the entire blood vessel as FeCl3 causes damage to the whole blood vessel, not only endothelial cells. In originally submitted figure 3, we measured the expression from the area where the CD31 and E-/P-selectin signals are merged. However, as the previously presented data could be interpreted as the expression of selectin specifically in endothelial cells, the expression of each molecule expressed in the entire blood vessel was modified and modified. The changes in the manuscript that are relevant to this explanation can be found in lines 146-157 (Results 3.3), 250-257 (Discussion). Additionally, references 31 and 32 were added.
3.
First, in verifying the antithrombotic efficacy of MF using FeCl3, we tried to reduce bias in drug administration as much as possible by using well-established experimental techniques. Therefore, intraperitoneal administration and oral administration were used. As can be seen in Figure 1, there was no difference in the efficacy of MF using either method. And the, this result was commissioned by Medicilion Preclinical Research and used to calculate PK parameters using Phoenix WinNonlin 7.0. We attempted to find out MF bioavailability when administered orally at 10 mg/kg. To calculate the bioactivity, the blood concentration value when the drug is injected or administered orally is required. The AUC value was derived based on this, and the bioavailability was determined to be 36.95 %.
Second, we have added the meaning of each parameter to the bottom of Table 3. The AUC was calculated based on the values listed in Table 2, and then the bioavailability was calculated. The sentence “When MF was injected ~ oral administration 4h,” previously on lines 207-208, was deleted, and it was judged that providing the values for each individual would provide more accurate results. Thus, these are indicated in Table 2 and Table 3. Also, an explanation has been added to lines 207-212 (Results 3.6) and lines 290 (Discussion).

Reviewer 2 Report
It is a very interesting and well written article.
Only few considerations:
-The background could be explained better and widely.
-Mumefural were can be found? Only in the Prunus mume? It is spread in the World? Or only in a place?
-This data could be generalised for all people?
Author Response
Thank you for your recommendations. We have highlighted the changes made in the revised manuscript in green. Mumefural, a citric acid derivative produced by the heated processing of Prunus mume or lemon juice, is widely used as a health food, especially in Japan (References 19 and 20). We have added this information to lines 39-43(Introduction). We have also added reference 20. In addition, this study verifies the antithrombotic efficacy of MF based on animal experiments, and a study on the human application of MF is planned for the future.
Reviewer 3 Report
Paper well done, well developed.
Paper on Mumefural (MF) is very clear in the study design and methods section. Minor Comments: In addition to the evaluation of platelet aggregation, it would also have been interesting to measure platelet activation markers such as serum thromboxane and plasma inflammatory markers to complete the vision of the antithrombotic effect of mumuferal.
Author Response
Thank you for your recommendations. The purpose of this study was to investigate the effect of MF on blood vessel tissue when inducing thrombus formation in animals. In the future, we plan to verify the efficacy of platelet aggregation and inflammatory markers of MF based on various biological samples such as serum, plasma, and proteins.
Round 2
Reviewer 1 Report
You will find below my comments to your answers according to the different points.
Point 1:
The corrections made by the authors are not sufficient and section 2.2 of Materials and methods is still not clear. The study design needs more clarification. How many groups with IP injection, with oral gavage? MF doses? For IP injection, what is the treatment duration? One injection only?
In section 2.3 data corresponding to figures 3 and 4 need explanation: how was the density calculated? What is the reference for this percentage? The thrombus area? Arterial lumen?
Point 2:
As the authors mentioned in their answer: CD31 antibody stained not only endothelial cells, but also other parts of blood vessels that underwent vascular damage with FeCl3. Thus, higher magnification may show at least luminal lining endothelial signal in control to validate labelling and allow the visualization of non specific signal in thrombus. As you suppressed images in fig 3, you can add that magnification to illustrate the labelling with the various antibodies and especially CD31.
Moreover, the decreased labellings observed in carotid of MF treated rats are certainly due to decreased thrombus formation and size rather than decreased protein expression. Indeed, platelets, macrophages and immune cells number is decreased in carotid artery lumen if thrombus is smaller. Thus fluorescence for E-selectin, P-selectin, ICAM, and VCAM labellings is also decreased. In that case, the decreased signal evaluated by decreased density does not reflect decreased protein expression. Otherwise, density calculation was not explained. To show decreased expression, a normalization to platelets, macrophages and immune cells numbers is required. Western blot allows that as generally total protein amount is used to normalize quantification. With those experiments, the authors have not shown modified cellular protein expression but local modified protein level due to decreased thrombus size.
Point 3:
Answer is ok. However, lines 288 and 290 the words:" … for a certain period…" and "…for a certain time …" are not scientifically relevant. The authors have to precise or suppress these words which mean nothing.
Author Response
Point 1:
The corrections made by the authors are not sufficient and section 2.2 of Materials and methods is still not clear. The study design needs more clarification. How many groups with IP injection, with oral gavage? MF doses? For IP injection, what is the treatment duration? One injection only?
In section 2.3 data corresponding to figures 3 and 4 need explanation: how was the density calculated? What is the reference for this percentage? The thrombus area? Arterial lumen?
Response:
Thank you for your recommendations. We are very honored that we can write a complete thesis, thanks to you. Following your comments, we have discussed a lot about the interpretation of the data. We have highlighted the changes made in the revised manuscript in yellow.
First, we added the description of the group based on drug treatment as follows:
“After 1 week of adaptation, the rats were randomly assigned to a total of 9 groups; 6 groups of varying drug dose by intraperitoneal injection (Control, FeCl3, 0.1, 1, 10 mg/kg of MF, and 10 mg/kg of Gingkolide [GA]) and 3 groups of varying drug dose by oral administration (Control, FeCl3, 10 mg/kg of MF).” (Lines 57-60)
“Doses of 0.1, 1, 10 mg/kg of MF, and 10 mg/kg of GA were administered intraperitoneally at once 30 min before FeCl3 treatment, while 10 mg/kg of MF was administered by oral gavage once a day for 1 week.” (Lines 63-65)
Second, the entire blood vessel was selected, and the red or green stained area was designated. The density was measured using ImageJ software and calculated as follows: Density of target molecule (%) = FeCl3 or drug treatment group / Control * 100 (%) (Please see lines 89-92).
Point 2:
As the authors mentioned in their answer: CD31 antibody stained not only endothelial cells, but also other parts of blood vessels that underwent vascular damage with FeCl3. Thus, higher magnification may show at least luminal lining endothelial signal in control to validate labelling and allow the visualization of non specific signal in thrombus. As you suppressed images in fig 3, you can add that magnification to illustrate the labelling with the various antibodies and especially CD31.
Moreover, the decreased labellings observed in carotid of MF treated rats are certainly due to decreased thrombus formation and size rather than decreased protein expression. Indeed, platelets, macrophages and immune cells number is decreased in carotid artery lumen if thrombus is smaller. Thus fluorescence for E-selectin, P-selectin, ICAM, and VCAM labellings is also decreased. In that case, the decreased signal evaluated by decreased density does not reflect decreased protein expression. Otherwise, density calculation was not explained. To show decreased expression, a normalization to platelets, macrophages and immune cells numbers is required. Western blot allows that as generally total protein amount is used to normalize quantification. With those experiments, the authors have not shown modified cellular protein expression but local modified protein level due to decreased thrombus size.
Response :
We investigated how MF affects the expressions of cell adhesion molecules involved in the process of thrombus formation in a FeCl3-induced thrombosis animal model. CD31 is a protein encoded in the PECAM-1 gene. It is a platelet/endothelial cell adhesion molecule and can be found not only in endothelial cells but also in various cells such as platelets, lymphocytes, and monocytes. Therefore, we tried to reduce confusion by changing the labeling of CD31 to PECAM-1 and presenting the results of platelet and cell adhesion molecules measurement.
In addition, since PECAM-1, E-selectin, P-selectin, ICAM, and VCAM all belong to the cell adhesion molecule family, Figure 3 and Figure 4 were combined. Thus, Figure 3 shows that MF inhibits thrombogenesis by reducing the increased expression of platelet/cell adhesion molecules in the FeCl3-induced thrombosis animal model. (Lines 172-178)
- Changed parts
- CD31 → PECAM-1 (all paper)
- Figure 4 → Figure 3D, 3E
- Figure 5 → Figure 4
- Figure 6 → Figure 5
Second, to observe changes in blood vessels as a method for verifying the efficacy of MF, it was judged that measuring the whole blood vessel picture was reasonable for data calculation. Accordingly, the density of each molecule in the blood vessel and the changed parts within the blood vessel was quantified, and it was judged that it was reasonable to present the whole blood vessel picture as a representative picture.
Point 3:
Answer is ok. However, lines 288 and 290 the words:" … for a certain period…" and "…for a certain time …" are not scientifically relevant. The authors have to precise or suppress these words which mean nothing.
Response:
We have deleted the phrase “and excreted for a certain period of time” for accuracy. As such, the sentences were modified as follows
“Through PK analysis, as reflected by Cmax (IV: 1594.54 ± 580.66 ng/mL, PO : 1731.61 ± 290.64 ng/mL) and Tmax (IV: 1 h, PO : 4 h), MF was present in the plasma after each single administration (IV: 2 mg/kg; oral: 10 mg/kg). Hence, the bioavailability of MF was found to be 36.95%.” (Lines 279-281).